# Intraindividual epigenetic heterogeneity underlying phenotypic subtypes of advanced prostate cancer

Kei Mizuno[1], Sheng-Yu Ku [1], Varadha Balaji Venkadakrishnan [1], Martin K. Bakht [1], Michael Sigouros [2], Joanna Chan[3], Anna Trigos [3,4,5,6], Jordan H. Driskill [7], Jyothi Manohar[2], Abigail King [2], Adam G. Presser[1], Min Jin Kim[1], Alok K. Tewari [1], Henry W. Long [8], David Quigley [9,10], Toni K. Choueiri [1], Steven Balk [11], Sarah Hill [1], Juan Miguel Mosquera [2,7], David Einstein[11], Shahneen Sandhu [4], Mary-Ellen Taplin[1] & Himisha Beltran [1] ✉

Castration-resistant prostate cancer is a heterogeneous disease with variable phenotypes commonly observed in later stages of the disease. These include cases that retain expression of luminal markers and those that lose hormone dependence and acquire neuroendocrine features. While there are distinct transcriptomic and epigenomic differences between castration-resistant adenocarcinoma and neuroendocrine prostate cancer, the extent of overlap and degree of diversity across tumor metastases in individual patients has not been fully characterized. Here we perform combined DNA methylation, RNA-sequencing, H3K27ac, and H3K27me3 profiling across metastatic lesions from patients with CRPC/NEPC. Integrative analyses identify DNA methylation-driven gene links based on location (H3K27ac, H3K27me3, promoters, gene bodies) pointing to mechanisms underlying dysregulation of genes involved in tumor lineage (ASCL1, AR) and therapeutic targets (PSMA, DLL3, STEAP1, B7-H3). Overall, these data highlight how integration of DNA methylation with RNA-sequencing and histone marks can inform intraindividual epigenetic heterogeneity and identify putative mechanisms driving transcriptional reprogramming in castration-resistant prostate cancer.

Tumor heterogeneity is a driver of treatment resistance in cancer and therefore represents a major obstacle for the development of precision medicine and targeted therapies. While prostate cancer typically arises as an androgen receptor (AR)-driven adenocarcinoma, reflecting its luminal prostate cell of origin[1], up to 15–20% of castration resistant prostate cancer (CRPC) tumors acquire alternative lineage programs as a mechanism of treatment resistance including transformation to a poorly differentiated small cell/neuroendocrine prostate cancer

[1]Department of Medical Oncology, Dana-Farber Cancer Institute, Harvard Medical School, Boston, MA, USA. [2]Englander Institute for Precision Medicine, Weill Cornell Medicine, New York, NY, USA. [3]Peter MacCallum Cancer Centre, Melbourne, VIC, Australia. [4]Sir Peter MacCallum Department of Oncology, University of Melbourne, Melbourne, VIC, Australia. [5]St. Vincent's Institute of Medical Research, Fitzroy, VIC, Australia. [6]Department of Biochemistry and Molecular Biology, Monash University, Melbourne, VIC, Australia. [7]Department of Pathology and Laboratory Medicine, Weill Cornell Medicine, New York, NY, USA. [8]Center for Functional Cancer Epigenetics, Dana-Farber Cancer Institute, Harvard Medical School, Boston, MA, USA. [9]Helen Diller Family Comprehensive Cancer Center, University of California San Francisco, San Francisco, CA, USA. [10]Department of Urology, University of California San Francisco, San Francisco, CA, USA. [11]Department of Medical Oncology, Beth Israel Deaconess Medical Center, Harvard Medical School, Boston, MA, USA. ✉e-mail: himisha_beltran@dfci.harvard.edu

(NEPC) with loss of AR-dependence and morphologic features often indistinguishable from small cell carcinomas arising from other sites[2-4]. Clinically, the diagnosis of NEPC is typically made through a single site metastatic biopsy, and patients with biopsy-confirmed NEPC histology are recommended to receive small cell carcinoma -directed therapies. However, there can be heterogeneity observed, sometimes with both NEPC and adenocarcinoma features present within tumors, and treatment efficacy can vary depending on the metastatic site and underlying phenotype.

The transcriptomic changes that distinguish castration-resistant NEPC from adenocarcinoma (CRPC-Adeno) are driven in part by underlying epigenetic alterations[2,5-9]. DNA methylation, histone marks, and chromatin accessibility are all epigenetic alterations, each contributing to transcriptional changes that can impact cell lineage and tumor phenotype. While it has been shown that global DNA methylation, H3K27ac and H3K27me3 histone patterns can begin to subtype phenotypic variants of CRPC[5,6,9-12], their extent of overlap and degree of diversity across metastases within and between patients has not been fully characterized.

Here, we perform comprehensive multi-omic profiling of metastatic CRPC tumors, including rapid autopsy cases with multiple anatomic sites from individual patients, to investigate the extent of epigenetic heterogeneity across metastases. We conduct integrated analyses of DNA methylation, RNA sequencing, and histone modifications (H3K27ac and H3K27me3) to understand the epigenetic mechanisms underlying phenotypic diversity. Additionally, we explore the regulatory networks driving tumor lineage programs and expression of therapeutic targets, with particular focus on cases exhibiting intraindividual heterogeneity.

## Results

### Patient-specific epigenetic signatures in metastatic CRPC
We evaluated 98 tumor tissue samples from 35 patients with metastatic CRPC (9 NEPC and 26 CRPC-Adeno); tissues from 21 patients were obtained at the time of rapid autopsy (8 NEPC, 13 CRPC-Adeno) where multiple anatomic sites were assessed (median 4 sites per patient) (Fig. 1a). Samples included lymph node ($n = 26$), liver ($n = 25$), lung ($n = 9$), bone ($n = 7$), prostate ($n = 8$), brain ($n = 9$), and other sites ($n = 14$). Tumor morphology was consistent in patients with multiple samples, meaning adenocarcinoma or NEPC morphology was conserved across metastases in these cases. Clinical features of the cohort are described in Supplementary Table 1. All samples underwent genome-wide DNA methylation by reduced representation bisulfite sequencing (RRBS)[13] and RNA-sequencing, and 35 samples also had matched ChIP-seq or CUT&Tag for H3K27ac and H3K27me3 marks based on tissue availability.

We first assessed genome-wide DNA methylation differences across all tumor samples using the top 10,000 most variably methylated CpG sites. We found that global methylation profiles were generally conserved across metastases within the same patient (Fig. 1b). This observed DNA methylation stability across metastases is consistent with a prior report from 2013 evaluating autopsy cases of CRPC-Adeno[14]. Hierarchical clustering identified the individuality of samples obtained from the same patient (Supplementary Fig. 1a). To evaluate the degree of similarity between samples, we computed Pearson correlation coefficients for pairs of samples from distinct patients and from the same patient revealing a significantly higher correlation coefficient when comparing samples from the same patient (Fig. 1c). This analysis also revealed the existence of outliers with notably low DNA methylation correlation even among samples from the same patient (specifically patients CA0090, WCM63, and WCM3672). Similarly, global RNA-seq profiles using the top 2,000 most variably expressed genes exhibited a conservation of overall gene expression profiles across metastases from the same individual (Fig. 1d, and Supplementary Fig. 1b) with a significantly higher correlation

coefficient (Fig. 1e). While not classified as outliers, samples from patients CA0090, WCM63, and WCM3672 also showed relatively low correlation coefficients in their RNA-seq profiles, appearing in the lower whisker region of the boxplot (Fig. 1e). We further investigated whether DNA methylation patterns were similar across samples from the same metastatic sites, regardless of the patient. Our analysis revealed that samples from the same anatomical site from different patients showed low correlation in methylation patterns. In contrast, high correlation coefficients were observed only when comparing samples from the same patient, regardless of the metastatic site. This finding suggests that DNA methylation patterns are more strongly influenced by patient-specific factors than by the site of metastatic spread (Supplementary Fig. 1c).

### Intraindividual phenotypic heterogeneity through molecular subtyping
Using previously defined AR signaling and NEPC signature gene sets that associate with CRPC-Adeno and NEPC subtypes[2,3,15], we performed clustering analysis and multidimensional scaling (MDS) on our gene expression data and identified 33 tumors as AR-positive (AR + )/neuroendocrine-negative (NE − ), 15 as AR-low/NE − , 8 as AR-negative(AR − )/NE − , 38 as AR − /neuroendocrine-positive (NE + ), and 4 as AR + /NE+ (Fig. 2a, and Supplementary Fig. 2a). These molecular subtypes are increasingly recognized as distinct biological states during prostate cancer progression and therapy resistance. The AR + /NE− subtype encompasses a typical, canonical AR-driven prostate adenocarcinoma, while AR-low/NE− reflects tumors with attenuated but still present AR signaling. The AR − /NE+ subtype corresponds to small cell/neuroendocrine prostate cancer with loss of AR expression and acquisition of neuroendocrine features. Notably, the AR + /NE+ (amphicrine) subtype represents tumors that co-express both AR and neuroendocrine programs, potentially capturing an intermediate state during lineage plasticity[15]. Less is known about the AR − /NE − (double-negative) subtype which could represent a distinct mechanism of AR-independence or develop along a continuum with NEPC as an intermediary state or later stage phenotype[16]. In our study, five patients exhibited more than one subtype (CA0090, DFCI24, DFCI53, WCM3672, WCM63), indicating intraindividual heterogeneity (Fig. 2a). To examine this intraindividual heterogeneity more thoroughly, we performed patient-specific clustering analyses for each of these five cases. For each patient, all samples were included, and clustering was based on the top 10,000 CpG sites showing the highest variance in DNA methylation β values within that patient's samples. This analysis revealed a clear separation between tumors of one subtype and those of another, suggesting that differences in DNA methylation underlie the molecular transitions observed in these cases (Fig. 2b, and Supplementary Fig. 2b).

### Gene expression regulation revealed by integrated epigenomic profiling
Next, we aimed to nominate which specific methylation changes impact gene expression and contribute to phenotypic diversity. We integrated DNA methylation and RNA-sequencing with histone profiling (see Methods). In general, DNA methylation involves the addition of a methyl group to cytosine, primarily occurring at CpG sites, and acts to suppress gene expression through recruiting repressor proteins, inhibiting transcription factor binding, and interacting with histones to modulate chromatin structure[17]. Acetylation of H3K27 is a modification of histone H3 and indicator of active enhancers, whereas trimethylation of H3K27 accumulates around gene promoter regions resulting in chromatin compaction and inhibition of gene expression[18,19].

We grouped CpG sites within 200 base pairs (bp) and selected clusters with three or more CpGs, yielding approximately 300,000 regions. These regions were classified into four categories based on

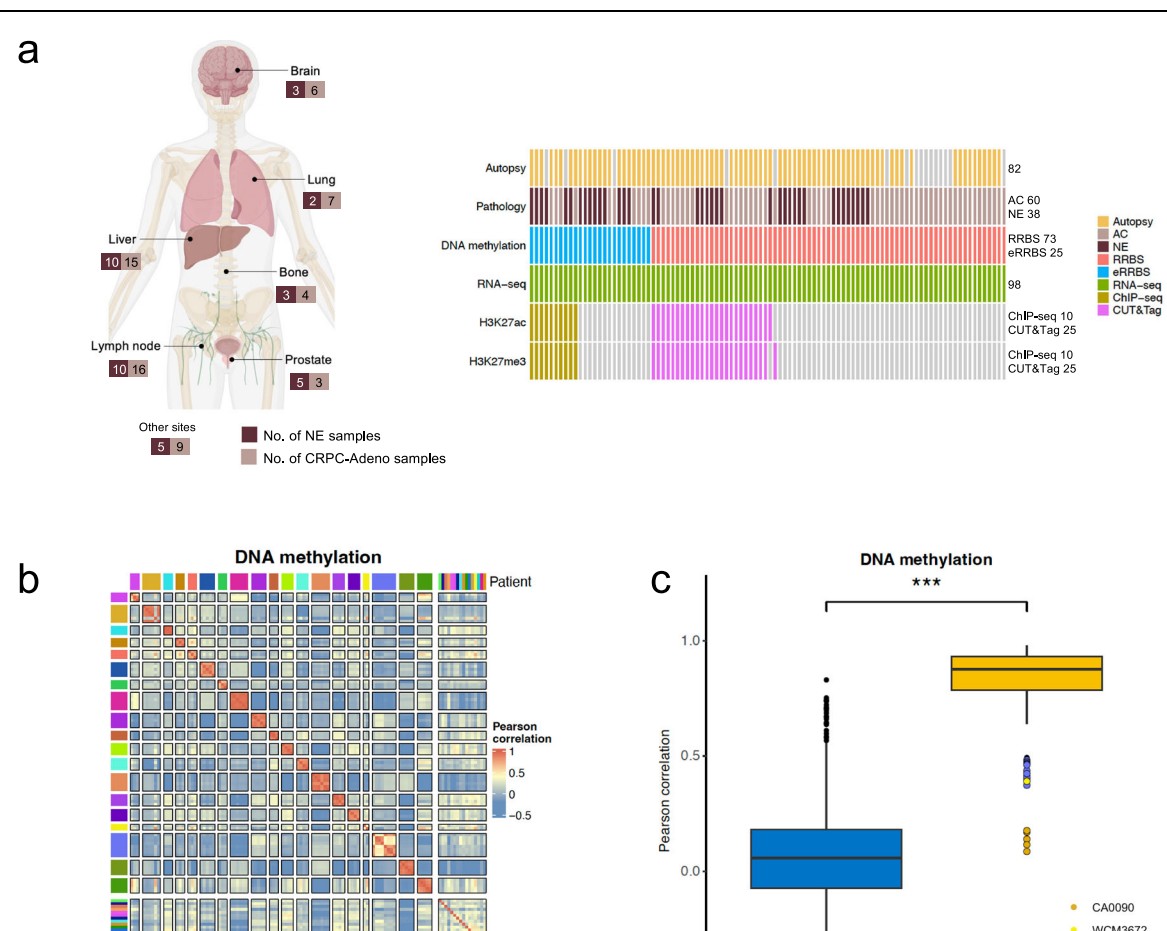

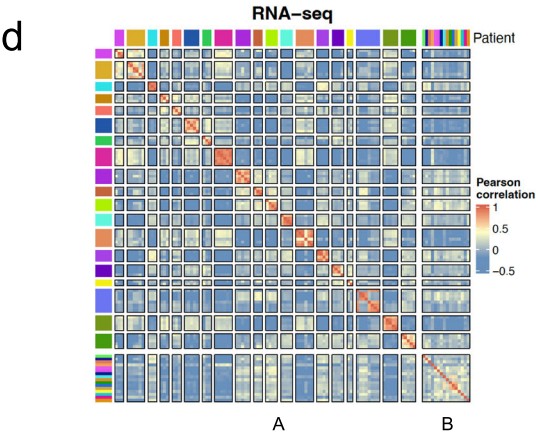

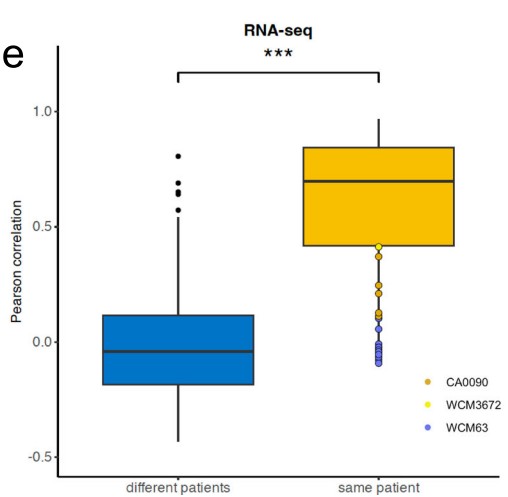

their location: H3K27ac-associated regions, H3K27me3-associated regions, promoter regions, or gene bodies. We computed the correlation between DNA methylation at each region and expression of its associated genes. Significant links were validated using a published dataset[6], resulting in 21,721 significant region-gene links involving 20,333 regions and 5707 genes (Fig. 3a, Supplementary Fig. 3a). Of the 20,333 regions, 19,418 (95.5%) were associated with only one gene and, on average, the expression of one gene correlated

with the methylation status of the two regions (Supplementary Fig. 3b). The majority of genes linked with H3K27ac-associated regions showed a negative correlation between their expression and DNA methylation levels (1640 out of 1968 genes, as shown in Fig. 3b). However, 70% of the genes linked with H3K27me3-associated regions exhibited positive correlation between their expression and methylation levels (507 out of 745 genes, Fig. 3b). This observation aligns with an earlier study showing that DNA methylation could be

**Fig. 1 | DNA methylation and transcriptional profiling of metastatic CRPC samples. a** Study design showing sample collection and multi-omic profiling strategy. Cases include 35 patients with metastatic CRPC (9 NEPC, 26 CRPC-Adeno), including 21 rapid autopsies with multiple anatomic sites assessed. Samples underwent DNA methylation profiling by RRBS/ERRBS, RNA sequencing, and H3K27ac/H3K27me3 ChIP-seq/CUT&Tag. NE, neuroendocrine. Figure 1/panel was partially created in BioRender. Mizuno, K. (2025) https://BioRender.com/hgg1f0e. **b** Correlation analysis of DNA methylation profiles. Panel A shows patients with multiple samples and panel B shows patients with single samples. Samples are grouped by patient and colored by patient ID. Color intensity represents Pearson correlation coefficients, demonstrating high correlation between samples from the same patient. **c** Pearson correlation coefficients of DNA methylation profiles between pairs of samples from the same patient (intra-patient, yellow, *n* = 156) versus different patients (inter-patient, blue, *n* = 4.597). Box plots show median, quartiles and whiskers extending to 1.5× the interquartile range. Data beyond the end of the whiskers are outliers and are plotted individually. ***, *p*-value < 0.001, statistical analyses used Wilcoxon two-sided tests. Patients with intraindividual heterogeneity (WCM63, CA0090, WCM3672) are highlighted. **d** Correlation analysis of gene expression profiles organized as in (**b**), showing high correlation between samples from the same patient. **e** Pearson correlation coefficients of gene expression profiles between intra-patient (yellow, *n* = 156) versus inter-patient (blue, *n* = 4.597) sample pairs. Box plots show median, quartiles and whiskers extending to 1.5× the interquartile range. Data beyond the end of the whiskers are outliers and are plotted individually. ***, *p*-value < 0.001, statistical analyses used Wilcoxon two-sided tests. Patients showing intraindividual heterogeneity in DNA methylation comparison (WCM63, CA0090, WCM3672) are highlighted.

inversely related to H3K27me3 enrichment[20], where high H3K27me3 signals correspond to hypomethylation and low gene expression, while low H3K27me3 signals correspond to hypermethylation and higher gene expression. As expected, the expression of most genes was negatively correlated with DNA methylation of their promoter regions. For gene bodies, there was no discernible pattern; the methylation level of a gene body region could be positively or negatively correlated with its expression (Fig. 3b).

For example, there was a region of the genome 7 kilobases upstream of the NE-lineage gene *ASCL1* that exhibited a strong positive correlation between *ASCL1* mRNA expression and H3K27ac signal in that area (Fig. 3c). Moreover, a negative correlation was observed between *ASCL1* expression and DNA methylation of that region (Fig. 3d). The neuronal factor *SEZ6L* had an H3K27me3-associated region in its gene body which showed a significant negative correlation between the H3K27me3 signal in the region and its expression (Fig. 3c). Additionally, there was a notable positive correlation between *SEZ6L* expression and DNA methylation levels in this region (Fig. 3d). There were significant negative correlations identified between *GSTP1* expression and methylation of its promoter region, and between *FOLH1* (PSMA) expression and the methylation level of its gene body (Fig. 3d). Overall, these data point to distinct epigenetic mechanisms of dysregulation of key prostate cancer biomarkers (*GSTP1*) including NE (*ASCL1, SEZ6L*), and luminal (e.g *FOLH1*) genes. Notably, when focusing on the *AR* gene, we identified two regions within the gene body exhibiting correlations between DNA methylation and gene expression (Supplementary Fig. 3c). However, the previously reported *AR* enhancer region[21–23] did not show significant correlations between DNA methylation and expression. In our cohort, a significant negative correlation between DNA methylation and *AR* gene expression was observed in the promoter region, but this finding was not replicated in the validation cohort (Supplementary Fig. 3d). We also evaluated patterns associated with other emerging cell surface targets in prostate cancer beyond PSMA. For *DLL3*, an inhibitory Notch ligand predominantly expressed in NEPC and other small cell carcinomas[24,25], a significant correlation between DNA methylation and gene expression was observed in two regions within the gene body (Supplementary Fig. 3e). For *STEAP1*, such correlations were found in regions 3 kb downstream and 46 kb downstream (Supplementary Fig. 3f). The *TROP2* gene (*TACSTD2*) showed significant correlation in the 3′ UTR region (Supplementary Fig. 3g). For the *B7-H3* gene (*CD276*), correlations were identified in one region each in the promoter and gene body (Supplementary Fig. 3h). Both *HER2* and *HER3* exhibited significant correlations between DNA methylation and gene expression in their respective promoter regions (Supplementary Fig. 3i). These findings suggest that epigenetic mechanisms may play a role in regulating the expression of therapeutic targets in prostate cancer, providing novel opportunities for refining biomarker assessment and personalizing treatment strategies potentially through modifying expression of the targets themselves.

## Characteristic pathway activation in double-negative CRPC

To further explore the molecular relevance of methylation-based gene links based on their location, we focused on tumors exhibiting intraindividual heterogeneity. We began by examining samples from WCM63, a patient with three AR + /NE− tumors and five tumors that lacked expression of both the AR and NE programs (AR − /NE − , or double negative, DN, CRPC). We queried differentially expressed genes (DEGs) between DN and AR + /NE− samples, which revealed 5800 upregulated and 3400 downregulated DEGs. We then extracted genes associated with the region-gene links detected, as well as genes whose associated regions showed more than a 10% methylation difference between the DN and AR + /NE− samples. These genes were nominated as potentially regulated by changes in DNA methylation. Among these genes, we further identified genes that were highly upregulated or downregulated in DN samples when compared across all samples. Highly upregulated genes were defined as those with expression levels above our defined threshold (mean+SD), while highly downregulated genes were defined as those with expression levels below the threshold (mean-SD). This analysis identified 421 highly upregulated and 319 highly downregulated genes in DN samples from WCM63 (Fig. 4a). Gene Ontology (GO) analysis for genes that were highly upregulated based on methylation of WCM63 DN samples pointed to the *BMP4* gene and the BMP4 signaling pathway (including *DLX3, LEF1, MSX2, WNT5A*) (Fig. 4b, Supplementary Table 2). We confirmed BMP4 expression at the protein level in the WCM63 DN sample (WCM63_Z11) by immunohistochemistry (Supplementary Fig. 4a). We calculated a BMP4 signaling signature score across all samples, including eight DN samples. Based on a previous report[26], we calculated this score as the sum of z-scores for the expression of eight genes (*BMP4, ID1, ID2, ID3, SMAD6, SMAD8, DLX3, SKIL*) revealing that BMP4 signaling was upregulated in 6/8 of the DN samples (5 from WCM63; 1 from WCM2343) (Fig. 4c). Analysis of published datasets of patient-derived xenograft (PDX) models, cell lines, and organoids[8,15,27,28] confirmed higher expression levels of BMP4 and target genes in DN tumors compared with other subtypes (AR + /NE − , AR-low/NE − , AR − /NE + , AR + /NE + ) (Supplementary Fig. 4b,c). We also analyzed other datasets[2,5,6] of CRPC patient samples and identified up to 35% of DN samples (6 out of 17 total DN samples) with elevated expression of BMP4 and genes associated with BMP4 pathway (Supplementary Fig. 4d,e). This included 5 out of 13 samples (38%) from the WCDT cohort and 1 out of 4 samples (25%) from the Beltran cohort, highlighting the heterogeneity in BMP4 pathway activation across different patient populations. There was not a specific enrichment of *BMP4* expression based on site of metastasis (Supplementary Fig. 4f). Bluemn et al[3]. previously reported FGF signaling as a driver of DN CRPC. Therefore, we also calculated an FGF signature score. The calculation method was similar to that of the BMP4 signature score, using the sum of z-scores for eight genes involved in FGF signaling (*FGF1, FGF8, FGF9, FGFBP1, FGFR1, FGFR2, FGFR3, FGFR4*). When comparing the FGF signature score with the BMP4 signature score, we found that samples with high

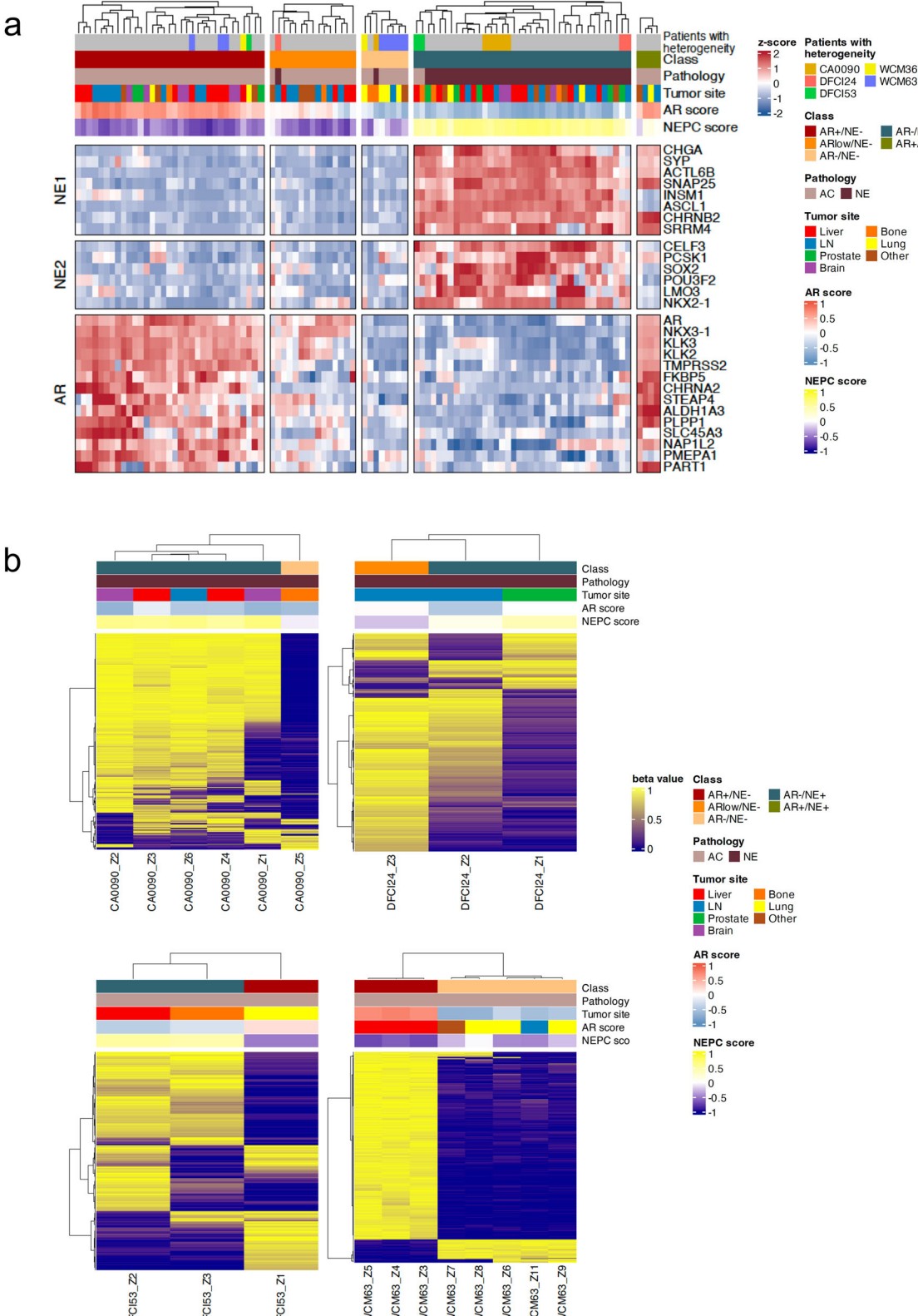

**Fig. 2 | Intraindividual heterogeneity in CRPC. a** Molecular subtype analysis (*n* = 98) reveals five cases with heterogeneous phenotypes across tumor sites (CA0090, DFCI24, DFCI53, WCM3672, WCM63). Heatmap shows the expression of genes associated with AR and NE program across samples. **b** DNA methylation clustering of samples from four cases showing intraindividual heterogeneity: CA0090 (*n* = 6), DFCI24 (*n* = 3), DFCI53 (*n* = 3), and WCM3672 (*n* = 8).

BMP4 signatures generally also exhibited high FGF signature scores (Supplementary Fig. 4g).

To functionally evaluate BMP4 signaling in DN tumors, we utilized the DU145 cell line, an established AR − /NE− cell line model. We treated DU145 cells with LDN-193189 (LDN), a small molecule inhibitor that selectively targets BMP4 receptor (ALK2/3) (Fig. 4d). Dose-response experiments revealed an IC50 of 2.1 µM ($R^2$ = 0.99) for LDN in DU145 cells (Supplementary Fig. 4h). Treatment with 2 µM LDN for

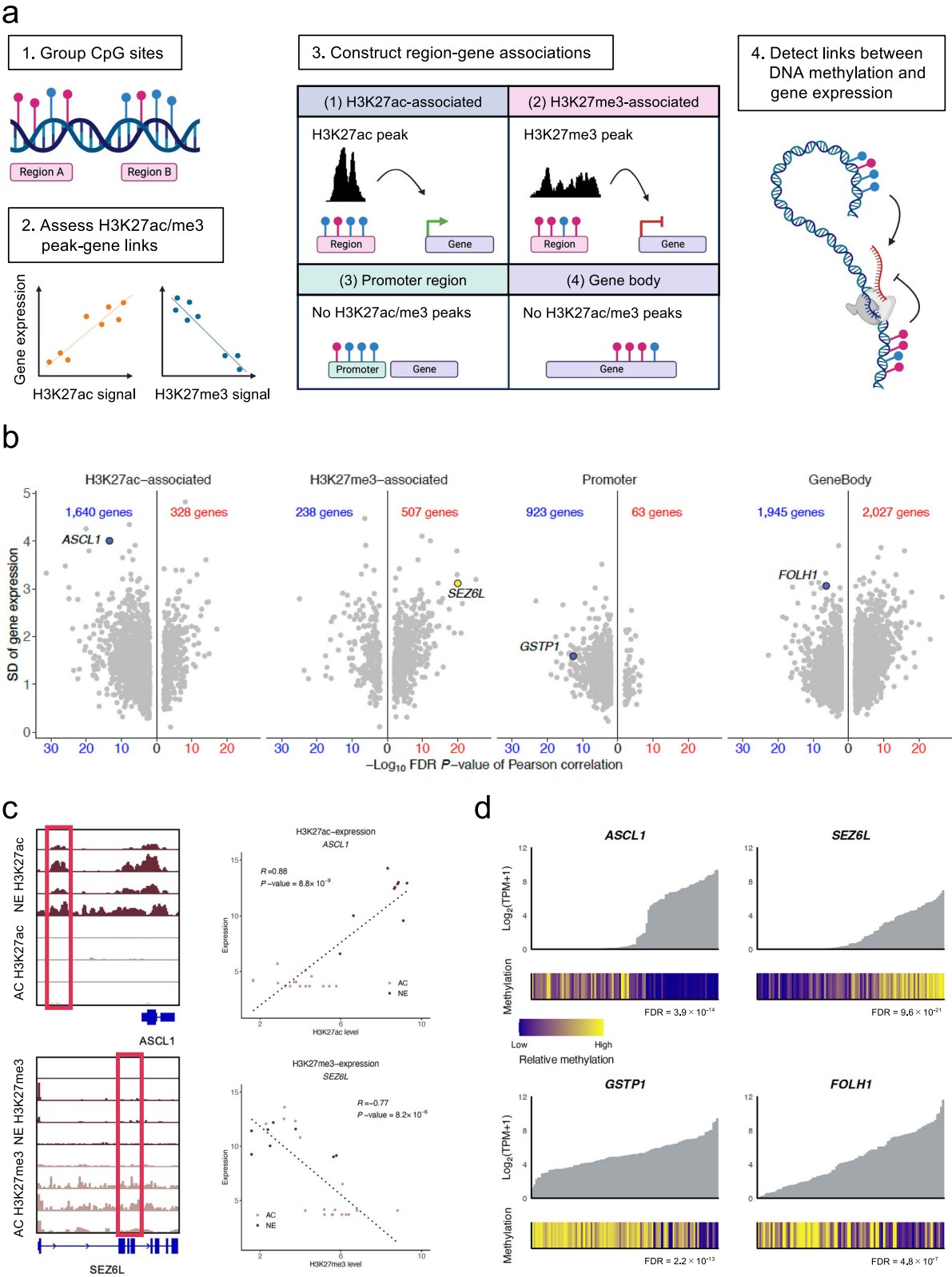

6 days resulted in a significant reduction in cell viability (68% decrease compared to vehicle control, $p < 0.001$) (Fig. 4e). To confirm that the observed effects were due to inhibition of BMP4 signaling, we assessed the phosphorylation status of SMAD1/5, a direct downstream target of BMP4 receptor activation (Fig. 4d). Western blot analysis demonstrated a marked reduction in phospho-SMAD1/5 levels following treatment with 2 μM LDN for 6 days, while total SMAD1/5 levels remained unchanged (Supplementary Fig. 4i). These findings suggest

that the BMP4 signaling pathway might contribute to the maintenance of cell viability in DNPC. To further explore the potential clinical relevance of BMP4 expression, we evaluated outcomes data from the West Coast Dream Team CRPC cohort[5,6], stratifying patients by BMP4 expression levels. Patients were categorized into high expression (TPM > 10, $n = 13$) and low expression (TPM ≤ 10, $n = 84$) groups. Kaplan-Meier analysis revealed no statistically significant difference in overall survival between these groups ($p = 0.32$, Supplementary

**Fig. 3 | Integration of DNA methylation, gene expression and histone modification data. a** Analytical workflow for identifying methylation-driven gene regulation. Figure 3/panel a was created in BioRender. Mizuno, K. (https://BioRender.com/703sb5t). **b** Distribution of correlations between DNA methylation and gene expression by genomic context (H3K27ac regions, H3K27me3 regions, promoters, gene bodies). Blue text indicates negative correlation between DNA methylation and gene expression, while red text indicates positive correlation. SD, standard deviation. **c** Example loci showing differential regulation. Left panels: Genomic regions (highlighted in red boxes) showing significant correlation between histone modifications and gene expression. Upper track shows region with positive correlation between H3K27ac signal and *ASCL1* expression; lower track shows region with negative correlation between H3K27me3 signal and *SEZ6L*

expression. Right panels: Scatter plots demonstrating these correlations between histone modification signals and gene expression. Upper panel: Scatter plot showing correlation between H3K27ac signal and *ASCL1* expression (AC: adenocarcinoma, pale pink, $n = 16$; NE: neuroendocrine, burgundy, $n = 9$). Lower panel: Scatter plot showing correlation between H3K27me3 signal and *SEZ6L* expression (AC, pale pink, $n = 17$; NE, burgundy, $n = 8$). Statistical significance was assessed using two-sided Pearson correlation tests. **d** Representative examples of methylation-expression relationships for key prostate cancer genes across samples ($n = 98$). Bar plots show sample-level gene expression values, with DNA methylation levels indicated by color. False discovery rate (FDR) values for correlations between gene expression and DNA methylation are shown.

Fig. 4j), though this assessment was limited by the relatively small sample size of patients with high BMP4 expression.

Notably, not all DN samples displayed this characteristic gene expression pattern. Further analysis of the methylation-regulated gene links of the two DN samples in our cohort (WCM3672 and CA0090) that did not exhibit upregulation of BMP4 or the pathway (Fig. 4c, f) revealed enrichment of GO terms related to cell-cell adhesion and immune processes (Fig. 4g, h; Supplementary Fig. 4k, l). There were 94 commonly methylation-upregulated genes between these two DN CRPC samples (Supplementary Fig. 4m). GO analysis of these genes revealed significant enrichment of immune-related processes, with several key immune regulatory genes including *VAV1*, *IL4R*, *CD300A*, *ITGB2*, and *CARD11* (Supplementary Table 3). Epigenetic changes associated with the upregulation of immune-regulatory genes, including those identified here, might contribute to the enhanced immune cell infiltration observed in this subset of DN CRPC. Immune scores were calculated for these sample based on the xCell software[29], which calculates cell type enrichment scores (ES) for 64 immune and stromal cell types. Among all samples in our study, these two DN samples displayed the highest immune scores (Fig. 4f). External cohorts of DN tumors with low BMP4 expression also displayed heterogeneous immune profiles. While this inverse relationship between BMP4 expression and immune scores was clearly observed in our cohorts, the WCDT cohort showed greater variability, with some DN samples exhibiting both low BMP4 expression and low immune scores (Supplementary Fig. 4d, e), suggesting that additional molecular drivers likely contribute to the development of this subtype.

## Discussion

Rapid autopsies provide a unique opportunity to evaluate multiple tissue specimens from individual patients to understand tumor heterogeneity and patterns of clonal evolution. Prior prostate cancer autopsy studies have supported a monoclonal origin of lethal prostate cancer, traceable back to an often multi-clonal primary prostate cancer, resulting in metastatic tumor cells maintaining a unique genomic signature while still accumulating new subclonal alterations with progression and treatment resistance[30]. Polyclonal seeding between CRPC metastases further leads to tumor heterogeneity and distinct patterns of alterations affecting the *AR* gene and tumor suppressors[31]. Tumor histology and transcriptomic studies have also pointed heterogeneity of phenotype in CRPC with a spectrum of morphologies and acquired transcriptional programs that are not seen in primary disease[15]. Here we build upon these important observations and layer on DNA methylation and histone distribution analyses of metastatic prostate cancer, showcasing how epigenetic changes contribute to tumor phenotype and transcriptional changes, mediating loss of expression of key luminal markers/targets such as the AR and PSMA and gain of new drivers such as ASCL1.

Our integrative approach combining DNA methylation with matched histone modification profiling (H3K27ac and H3K27me3) across multiple samples within the same cohort provides unique insights into the epigenetic regulation of gene expression in patients with CRPC.

While previous studies[5,6] have examined correlation between DNA methylation and gene expression in advanced prostate cancer, prior methylation analyses were largely confined to promoter and gene body regions or relied on external data to identify putative enhancers. Our comprehensive multi-omic approach enabled the identification of over 20,000 regions where DNA methylation changes likely contribute to altered gene expression, offering a more complete picture of the epigenetic landscape. Our approach uncovered the possibility of DNA methylation regulation of clinically relevant cell-surface targets, including PSMA, DLL3, and B7-H3, suggesting that epigenetic modifications play a crucial role in regulating therapeutic target expression. These findings could be translated into clinical practice in the future through the analysis of circulating cell-free DNA (cfDNA) methylation patterns. The general stability of epigenetic changes across metastatic tissues that we observed further supports the potential utility of cfDNA methylation profiling as a readily available dynamic biomarker. This approach could enable real-time monitoring of tumor molecular phenotype, including the development of lineage plasticity, and help infer expression of cell surface targets to guide the timing of targeted therapies such as antibody-drug conjugates (ADCs), radionuclides, T cell engagers, or other approaches. The potential of cfDNA-based epigenetic assays is increasingly being refined and extended through integration with other data types such as cfDNA histone marks and chromatin accessibility[32,33], which could provide even more comprehensive insights into tumor evolution and inform therapeutic decision-making.

Our integrated analysis also revealed distinct molecular features of DN CRPC, with a subset showing elevated BMP4 signaling pathway activation and another exhibiting higher immune cell infiltration. The BMP4 pathway has been implicated as a regulator of cellular differentiation during embryonic growth and cancer development; in cancer, this pathway can contribute to tumor progression, inducing cell proliferation, and epithelial-mesenchymal transition (EMT), and enhance cancer cell survival by inhibiting apoptosis[34,35]. Our data suggests that epigenetic alterations may underlie the phenotypic switch from AR + /NE− to DN CRPC associated with upregulation of genes involved in BMP4 signaling. A prior report had described patients with high *BMP4* expression in primary tumors having a higher frequency of bone metastases (28%) compared to those with low expression (19%)[36]. In a mouse xenograft model using breast cancer cells, BMP4 treatment led to an increase in the number of bone metastases, suggesting that BMP4 could have a potential role in promoting bone metastasis[37]. However, in our cohort, we did not observe a specific enrichment of *BMP4* expression in bone metastasis samples. We found that BMP4 and FGF pathways were both upregulated in DN CRPC. While the cross-talk between BMP signaling and the FGF pathway in cancer remains unclear, studies using embryonic stem cells have revealed that BMP signaling is a crucial inducer of the totipotent state; however, its role is constrained by the cross-activation of multiple pathways, including FGF[38]. This interplay between BMP and FGF can contribute to stem cell heterogeneity and fate determination, suggesting that similar mechanisms might be at play in cancer cell

populations. Regarding the therapeutic approach for DN CRPC tumors exhibiting BMP4 pathway activation, our functional experiments suggest that targeting this pathway might be promising and further studies are warranted. Although we did not observe a statistically significant correlation between BMP4 expression levels and overall survival, this finding should be interpreted with caution due to the limited sample size, particularly in the high expression group. Larger studies will be necessary to definitively establish whether BMP4 expression has independent prognostic value in advanced prostate cancer.

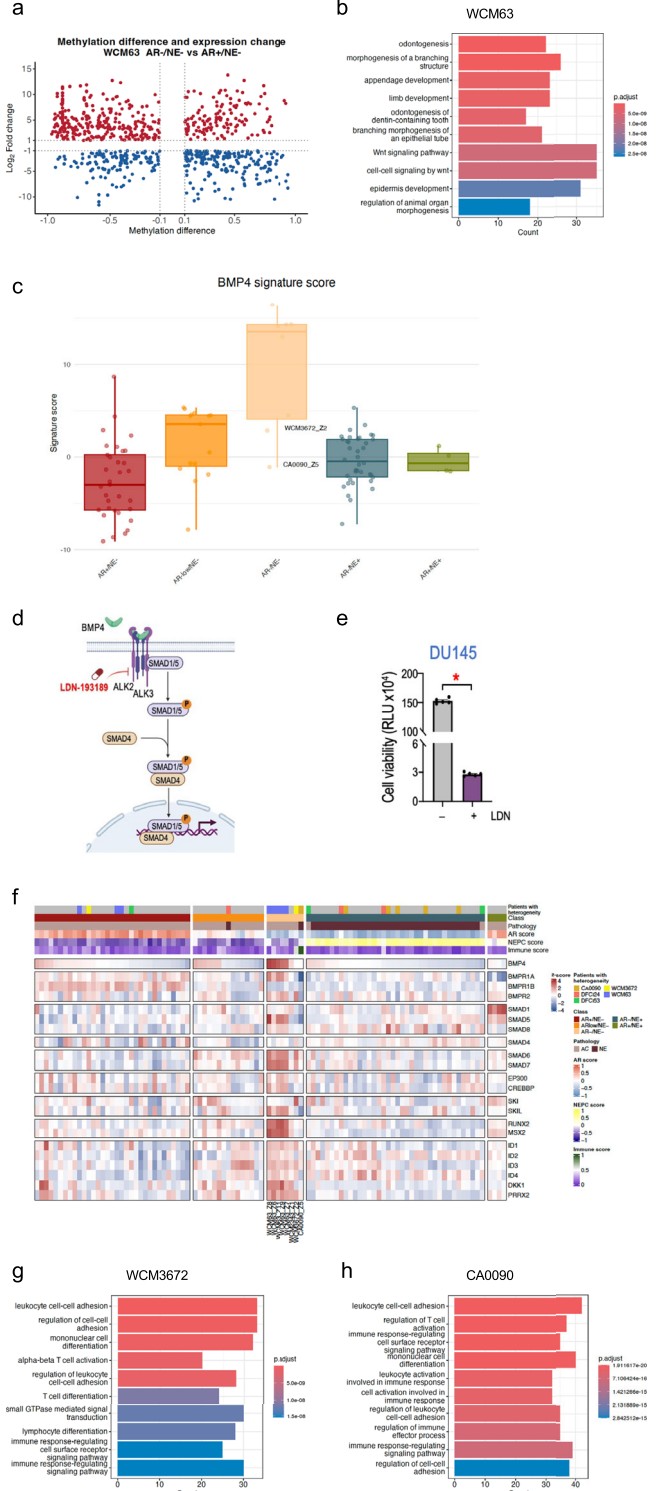

**Fig. 4 | BMP4 pathway and immune signatures define distinct double negative CRPC. a** DNA methylation-regulated genes identified in WCM63 AR − /NE− versus AR + /NE− samples. Red and blue dots indicate highly upregulated and downregulated genes, respectively, that show significant expression changes and substantial methylation differences in their associated regions. **b** Gene Ontology analysis of identified DNA methylation-upregulated genes in WCM63 double negative samples ($n = 5$) shows enrichment of GO terms with BMP4 and its downstream targets. Statistical analysis for GO enrichment was performed using one-sided Fisher's exact test with Benjamini-Hochberg correction for multiple testing. **c** BMP4 signature scores across all sample subtypes: AR + /NE− ($n = 33$, red), AR-low/NE− ($n = 15$, orange), AR − /NE− ($n = 8$, light orange), AR − /NE+ ($n = 38$, green), and AR + /NE+ ($n = 4$, blue). The plot shows elevated pathway activity in most double negative samples, with low scores in double negative samples from WCM3672 and CA0090. Box plots show median, quartiles and whiskers extending to 1.5× the interquartile range. Individual data points are plotted as overlaid dots. **d** Schematic of BMP4 signaling and ALK inhibition using LDN-193189. Figure 4/panel d was created in BioRender. Venkadakrishnan, VB. (https://BioRender.com/my372kw). **e** DU145 cells treated with vehicle (−) or 2 μM LDN-193189 (LDN, ALK inhibitor) for 6 days and assessed using CellTiter-Glo® luminescent cell viability assay. Data represent mean values ± SEM from five independent biological replicates per condition ($n = 5$ independent experiments/condition). Gray columns, vehicle treatment; purple columns, LDN-193189 treatment. Statistical significance was determined using Wilcoxon two-sided tests; *, $p < 0.05$; RLU, Relative Light Units. **f** Heatmap showing expression of BMP4 and BMP4 signaling pathway-related genes across all samples ($n = 98$), with corresponding immune scores shown. Double negative samples ($n = 8$) exhibit either high BMP4 pathway activity or high immune scores in this cohort. **g, h** Gene Ontology analysis of DNA methylation-upregulated genes in double negative samples from WCM3672 (**g**) and CA0090 (**h**) reveals enrichment of immune-related processes. Statistical analysis for GO enrichment was performed using one-sided Fisher's exact test with Benjamini-Hochberg correction for multiple testing.

The identification of DN CRPC samples with high immune scores is particularly intriguing in light of recent evidence demonstrating the critical role of DNA methylation in shaping the tumor immune microenvironment[39]. DNA methylation has been shown to regulate immune cell infiltration and function, with implications for both tumor progression and therapeutic response[40]. The distinct epigenetic profiles we observed in immune-high DN CRPC suggest potential opportunities for therapeutic intervention, as epigenetic therapies have demonstrated synergistic effects with immunotherapy in preclinical studies[41]. This approach is supported by data from other cancer types, such as azacitidine plus pembrolizumab in myelodysplastic syndrome[42], and guadecitabine plus ipilimumab in advanced melanoma[43]. Further characterization of the immune landscape through single-cell analysis and spatial transcriptomics would help provide insights into the composition and spatial organization of immune cells within these tumors.

When examining the potential influence of prior therapies on tumor heterogeneity in our cohort, there was no clear correlation between the number of systemic treatments and the presence or degree of intraindividual phenotypic heterogeneity. Amongst patients exhibiting heterogeneity across metastatic sites, treatment histories varied considerably. For instance, patient WCM63, who demonstrated distinct AR + /NE− and AR − /NE− phenotypes across different metastases, had received eight different systemic therapies prior to autopsy. In contrast, other cases with heterogeneous phenotypes (CA0090, DFCI24, WCM3672) had undergone only 2-3 types of prior systemic therapy. This observation suggests that mechanisms beyond cumulative treatment exposure are likely driving phenotypic divergence. The epigenetic rewiring documented in this study indicates that intrinsic tumor plasticity and epigenome-mediated adaptations may play more significant roles in determining phenotypic fate during disease progression than the specific sequence or quantity of therapeutic interventions. Understanding these complex epigenetic

mechanisms of adaptation may ultimately inform more effective treatment sequencing and combination strategies to address heterogeneity in advanced prostate cancer.

Our study remains with limitations due to the relatively small sample size and limited tumor specimens to perform additional complementary analyses. Validation across multiple external datasets suggests these findings may have broader biological relevance. However, the heterogeneity observed even within DNPC subtype highlights the need for more extensive sampling to fully characterize the spectrum of epigenetic mechanisms driving lineage plasticity in advanced prostate cancer. Additionally, while our study focused on H3K27ac and H3K27me3 due to their complementary roles in gene regulation and particular relevance to prostate cancer phenotypic transitions[9], future investigations with greater tissue availability could incorporate additional histone modifications such as H3K4me1, H3K4me3, and H3K9me3 to further refine our understanding of the complex epigenetic landscape driving tumor heterogeneity. However, we hope this unique dataset will be important resource for the community and helps inform future studies focused on multi-omic data integration including with genomic information, functional characterization of epigenetic reprogramming and crosstalk in prostate cancer, and the development of epigenetic therapies or co-targeting approaches based on tumor phenotype.

## Methods

### Sample collection
Tumor specimens were collected through Institutional Review Board (IRB)/ethics approved protocols at DFCI (20-000, 19-883), WCM (1305013903), Beth Israel Deaconess (15-441) and Peter MacCallum Cancer Center (HREC/15/PMCC/56) after written informed consent. Tumors histology was reviewed and classified as adenocarcinoma or NEPC based on morphology[44] by a board-certified pathologist. Representative histological images (hematoxylin and eosin (H&E) staining) from different molecular subtypes identified in our cohort are provided in Supplementary Fig. 5. Punches were made in frozen OCT section and processed for RNAseq, RRBS, and CUT&Tag.

### Sequencing and data processing
**Reduced representation bisulfite sequencing (RRBS).** The procurement of genomic DNA from tissue samples was achieved by the application of the restriction enzyme MspI, followed by bisulfite conversion. RRBS/ERRBS libraries were prepared as previously described[13]. The resulting DNA was subsequently amplified by PCR and libraries were prepared for Illumina sequencing. Raw sequencing data were processed using the Bismark[45] pipeline, which encompassed quality control, alignment with the GRCh37 reference genome, and methylation calling.

**RNA sequencing (RNA-seq).** mRNA was obtained from the tissue samples, and libraries were established using the Illumina TruSeq Stranded mRNA Library Prep Kit. Sequencing was performed on the Illumina platform, resulting in paired-end reads. The raw sequencing data were then processed using the STAR[46] aligner for mapping to the human reference genome, followed by quantification of gene expression using RSEM[47].

**Chromatin immunoprecipitation sequencing (ChIP-seq).** Chromatin immunoprecipitation sequencing (ChIP-seq) was carried out on 10 samples to investigate histone modifications, specifically the H3K27ac and H3K27me3 modifications. Chromatin was isolated from the tissue samples, and immunoprecipitation was performed using antibodies specific to H3K27ac and H3K27me3. Immunoprecipitated DNA was then prepared for sequencing on an Illumina platform. Raw sequencing data were processed using the MACS2[48] pipeline for peak

calling. We used the "-f BAM" flag with a genome size of 2.7e9 and the standard FDR threshold of 0.01. For H3K27me3 data, the flag "--broad" was used.

**Cleavage under targets and tagmentation (CUT&Tag).** CUT&Tag was utilized as an alternative technique to ChIP-seq to profile H3K27ac and H3K27me3 histone modifications in 25 samples each. This methodology entails the use of antibody-targeted micrococcal nucleases to cleave chromatin, which is subsequently followed by tagmentation and PCR amplification. The resulting libraries were sequenced on the Illumina platform and the data were processed using the MACS2 pipeline for peak calling. We used the "-f BAMPE" flag with a genome size of 2.7e9 and the standard FDR threshold of 0.01. For H3K27me3 data, the flag "--broad" was used.

### Identification of global methylation patterns and transcriptomic patterns
All DNA methylation information was collapsed to the forward strand. We extracted CpG sites that were covered across all samples, resulting in a total of 1,939,425 CpG sites. From these CpG sites, we calculated β values for bases with coverage ≥5 in all samples. For downstream analysis, we selected 10,000 CpG sites showing the highest variance in β values across samples and performed hierarchical clustering. To assess methylation pattern similarities between samples, we calculated Pearson correlation coefficients using the β values of these CpG sites. We then compared the correlation coefficients between samples derived from the same patient versus samples from different patients using the Wilcoxon rank sum test to determine if there were statistically significant differences in methylation patterns based on patient origin.

For transcriptome analysis, read counts were normalized using DESeq2[49]. Genes with TPM values < 1 across all samples were excluded from further analysis. The top 2000 genes showing the highest expression variance were selected for clustering analysis. Correlation analysis between samples was performed using Pearson's correlation coefficient, following the same approach used in the DNA methylation analysis.

### Cluster analysis using AR and NE expression
Samples were clustered based on the expression of previously reported AR and NE signature genes[3,6]. For each sample, AR and NEPC scores were calculated using the previously described method[3] with the GSVA Bioconductor package[50] in R. Additionally, we calculated z-scores for each gene in the respective signature sets across all samples. Samples were classified as AR-negative (AR − ) if they had an AR score < 0 and the sum of *AR* and *KLK3* expression z-scores was < 0; otherwise, they were classified as AR-positive. Within AR-positive samples, those with a sum of z-scores for the AR signature gene set > 3.5 were classified as AR + , while those with ≤3.5 were classified as AR-low. Similarly, samples with a sum of z-scores > 0 for the NE1 gene set (shown in Fig. 2a) were classified as NE + ; otherwise, they were classified as NE − . Using these criteria, we identified molecular subtypes (AR + /NE − , AR-low/NE − , AR − /NE − , AR − /NE + , and AR + /NE + ). To visualize the relationships between these phenotypic groups, we employed classical multidimensional scaling (MDS) using the cmdscale function in R, based on the expression profiles of the gene set.

### DNA methylation analysis of specific cases with intraindividual heterogeneity
For five patients showing intraindividual heterogeneity (CA0090, DFCI24, DFCI53, WCM3672, WCM63), clustering analysis was performed for each patient using the top 10,000 CpG sites showing the highest variance in β values among samples from the same patient. To

examine genome-wide DNA methylation patterns across molecular subtypes, we divided the genome into 100,000 bp bins and calculated the mean β values per bin for each molecular subtype.

## Inference of DNA methylation-gene expression regulatory chains

After the identification of H3K27ac narrow peaks and H3K27me3 broad peaks from the ChIP-seq and CUT&Tag data, a union set of peaks was created for each histone modification. For each sample with CUT&Tag data, fragment counts within each histone peak were extracted using the getCounts function from chromVAR[51], followed by variance-stabilizing transformation (vst) normalization using DESeq2.

Using these normalized fragment counts and gene expression data, we calculated Pearson's correlations between histone peaks and expression of genes with transcription start sites (TSS) within ±0.5 Mb of the peaks. Correlations were considered significant if they met FDR < 0.01 and showed positive correlation between H3K27ac peaks and gene expression, or negative correlation between H3K27me3 peaks and gene expression. Peaks within 1000 bp of each other that showed significant correlations with the expression of the same gene were merged. Peaks that were not detected in at least two samples were excluded from further analysis.

To establish links between DNA methylation and gene expression, we clustered CpG sites within 200 bp and selected regions with three or more CpGs. These regions were classified into different categories based on their genomic location and association with histone modifications. Regions within ±1 kb of previously identified H3K27ac peaks with significant gene expression correlations were classified as "H3K27ac-associated" regions. Similarly, regions within ±1 kb of H3K27me3 peaks with significant gene expression correlations were classified as "H3K27me3-associated" regions. Regions in promoter regions (1,500 bp upstream of TSS) or gene bodies were extracted regardless of proximity to H3K27ac or H3K27me3 peaks and classified as "promoter region" or "gene body" respectively. For H3K27ac-associated and H3K27me3-associated regions, we performed additional annotation using annotatr[52] and assigned them to genomic features in the following hierarchical order: promoter, 5'UTR, 3'UTR, gene body. Finally, we analyzed correlations between the methylation β values of each region and the expression of their associated genes. For H3K27ac-associated and H3K27me3-associated regions, the associated genes were those showing significant expression correlation with nearby histone modification peaks. For promoter regions and gene bodies, the associated genes were those containing or adjacent to the respective regions. Correlations with FDR < 0.01 were considered significant. Significant links were validated using the WCDT dataset[5,6].

## Identification of DNA methylation-regulated genes

To identify genes potentially regulated by DNA methylation changes, we first identified differentially expressed genes (DEGs) between different tumor subtypes (AR − /NE− vs. AR + /NE− ) in WCM63 samples using DESeq2. Genes with |log2(fold change (FC))| > 1 and Benjamini-Hochberg (BH) adjusted $p$-value ≤ 0.05 were considered significantly differentially expressed. We then extracted the genes associated with the region-gene links established in the previous step, revealing the DEGs that could potentially be regulated by DNA methylation changes. Among these genes, we further extracted genes that were highly upregulated or downregulated in AR − /NE− tumor subtype when compared across all samples. Highly upregulated genes were defined as those with expression levels above the mean+SD, whereas highly downregulated genes were defined as those with expression levels below the mean−SD. We confirmed that their associated regions exhibited more than a 10% methylation difference compared with AR + /NE− tumor subtype. For WCM3672 and CA0090, which each contained only one AR − /NE− sample, we modified our approach to identify DNA methylation-regulated genes. We compared gene

expression between the AR − /NE− sample and their respective reference tumor subtypes (AR + /NE− for WCM3672 and AR − /NE+ for CA0090). Genes with |log2(FC)| > 2 were defined as DEGs. Similar to the WCM63 analysis, we identified highly upregulated and downregulated genes as those with expression levels above mean+SD or below mean −SD, respectively. Among these genes, we selected those with associated regions showing more than 10% methylation difference between subtypes. GO analysis was performed on the identified genes using the enrichGO function from the clusterProfiler[53] package. Terms with BH adjusted p-value ≤ 0.05 were considered significantly enriched.

## Validation experiments

**Cell viability assays.** DU145 cells (HTB-81) were obtained from the American Type Culture Collection (ATCC), and their authentication was confirmed through short tandem repeat (STR) profiling. Cells were routinely tested for mycoplasma contamination every six months using a mycoplasma PCR detection kit (ABM; cat. no. G238). These cells were seeded at a density of 2000 cells per well in a 96-well plate and treated with vehicle (water) or LDN-193189 (Selleckchem, cat. no. S7507) at concentrations as indicated in the figure legends after 24 hours of seeding. Cell viability was measured after 6 days of treatment using CellTiter-Glo assay (Promega) in 96-well plate format as per the manufacturer's protocol.

**Western blotting.** Cells were washed twice with ice-cold phosphate-buffered saline (PBS, Gibco) and harvested in whole-cell lysis buffer (110 mM sodium dodecyl sulfate (SDS), 100 mM dithiothreitol, 80 mM Tris-HCl (pH 6.9), 10% glycerol). Cell lysates were boiled at 95 °C for 5 min. Equal amounts of protein was estimated by Lowry protocol (RCDC Protein Assay, Bio-Rad) and were subjected to NuPAGE Novex gel electrophoresis (Life Technologies). Based on the predicted molecular weight of the protein of interest, Gel electrophoresis was performed on 4–12% or 10% Bis-Tris NuPAGE gels or 3–8% Tris-Acetate NuPAGE gels. Proteins were transferred onto nitrocellulose membranes (NuPAGE) and blocked with 5% bovine serum albumin prepared in tris-buffered saline with 0.1% Tween-20. Primary antibodies against Phospho-SMAD1/5 (Ser463/465) (cell signaling technology; cat. no. 9516; 1:1000 dilution), Total SMAD5 (cell signaling technology; cat. no. 12534; 1:1000 dilution), or β-actin (cell signaling technology; cat. no. 3700S; 1:5000 dilution) were used as indicated in the figures. Near Infra-Red Fluorescent detection was performed using LI-COR secondary antibodies matching the host species of the primary antibodies. LI-COR Odyssey M and imaging software were used to visualize the western blots. Blots were stripped with stripping buffer (Restore Plus Western blot, ThermoFisher Scientific) and reprobed with additional primary antibodies as required.

**Immunohistochemistry (IHC).** IHC was performed on the Leica Bond III automated staining platform using the Leica Biosystems Refine Detection Kit (Leica; DS9800). FFPE tissue sections were baked for 30 minutes at 60 °C and deparaffinized (Leica AR9222) prior to staining. Anti-BMP4 antibody (antibodies.com, catalog number A80532, polyclonal) was used at 1:50 dilution following 20-minute EDTA antigen retrieval (Leica ER2 AR9640). The primary antibody was incubated for 30 minutes, visualized via DAB, and counterstained with hematoxylin (Leica DS9800). The slides were rehydrated in graded alcohol and coverslipped using the HistoreCore Spectra CV mounting medium (Leica 3801733).

## Statistical analysis

Statistical analyses were performed using R software (version 4.2). For comparisons between two groups, we employed the Wilcoxon rank-sum test as a non-parametric method to assess statistical significance. To evaluate relationships between variables, Pearson correlation coefficients were calculated to quantify the strength and direction of

linear associations. Statistical significance was defined as $p < 0.05$ for all analyses. Graphical representations were generated using the ggplot2 package in R. No statistical method was used to predetermine sample size. No data were excluded from the analyses. The experiments were not randomized. The Investigators were not blinded to allocation during experiments and outcome assessment.

## Reporting summary

Further information on research design is available in the Nature Portfolio Reporting Summary linked to this article.

## Data availability

RRBS, RNA-seq, ChIP-seq and CUT&Tag for H3K27ac and H3K27me3 data of patient tumor samples generated in this study have been deposited in the public repository (Gene Expression Omnibus) under accession codes GSE289466, GSE289467, GSE289468, and GSE289605. The RNA-seq data of mCRPC tumors from the WCDT cohort are available in the European Genome-Phenome Archive database under accession codes EGAD00001008991, EGAD00001008487, and EGAD00001009065[5,6]. The publicly available RNA-seq data of LuCaP and other PDX models, cell lines, and organoids are available in the Gene Expression Omnibus database under accession codes GSE126078[15], GSE160393[27], GSE199596[28], and GSE199190[8]. The publicly available methylation data of mCRPC tumors from the WCDT cohort are available in the dbGaP database under accession codes phs001648 [https://www.ncbi.nlm.nih.gov/projects/gap/cgi-bin/study.cgi?study_id=phs001648.v2.p1][5] and in the European Genome-Phenome Archive database under accession codes EGAS00001006649[5,6]. The methylation data from LuCaP PDX models are available in the Gene Expression Omnibus database under accession codes GSE227853 [https://www.ncbi.nlm.nih.gov/geo/query/acc.cgi?acc=GSE227853][54]. All raw data used to generate the figures presented in this manuscript are available in the Source Data file. Source data are provided with this paper.

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

## Acknowledgements

Some figures were created using BioRender.com. The super-computing resources were provided by Human Genome Center, Institute of Medical Science, the University of Tokyo. We would like to thank Ahmed Elsaeed for pathology assistance. We would like to thank Dana-Farber/Harvard Cancer Center in Boston, MA, for the use of the Specialized Histopathology Core, which provided histology and immunohistochemistry service. Dana-Farber/Harvard Cancer Center is supported in part by an NCI Cancer Center Support Grant # NIH 5 P30 CA06516. K.M. was supported by the Japan Society for the Promotion of Science. V.B.V. is supported by DoD PCRP Early Career Investigator Award (W81XWH2210197), National Cancer Center Postdoctoral Fellowship Award, and Prostate Cancer Foundation Young Investigator Award (23YOUN15). T.K.C is supported in part by the Dana-Farber/Harvard Cancer Center Kidney SPORE (2P50CA101942-16) and Program 5P30CA006516-56, the Kohlberg Chair at Harvard Medical School and the Trust Family, Michael Brigham, Pan Mass Challenge, Hinda and Arthur Marcus Fund and Loker Pinard Funds for Kidney Cancer Research at DFCI. D.E is supported by the Prostate Cancer Foundation Young Investigator Award and the Dana-Farber/Harvard Cancer Center Prostate SPORE (P50 CA254861-01A1). H.B. is supported by the Prostate Cancer Foundation, DoD PCRP (W81XWH-17-1-0653, HT94252310407) and NIH/NCI (R37CA241486-01A1, WCM SPORE P50 CA211024-01A1), DF/HCC SPORE P50 CA272390-01.

## Author contributions

Conceptualization: K.M., H.B. Methodology: K.M., S.Y.K., V.B.V. Data Curation: K.M., M.S., J.C., J.M., A.K. Formal Analysisdas: K.M., A.T. Investigation: K.M., S.Y.K., V.B.V., M.K.B., J.H.D., A.G.P., M.J.K. Resources: H.W.L., D.Q., S.H., J.M.M., D.E., S.S., M.E.T., H.B. Software: K.M. Validation: K.M., S.Y.K., V.B.V., J.H.D., A.G.P., M.J.K., D.Q. Writing – Original Draft: K.M., H.B. Writing – Review & Editing: K.M., S.Y.K., V.B.V., M.K.B., M.S., J.C., A.T., J.H.D., J.M., A.K., A.G.P., M.J.K., A.K.T, H.W.L., D.Q., T.K.C., S.B., S.H., J.M.M., D.E., S.S., M.E.T., H.B. Visualization: K.M., V.B.V., A.G.P. Supervision: A.K.T., H.W.L., T.K.C., S.B., J.M.M., S.S., M.E.T., H.B. Project administration: K.M., H.B. Funding acquisition: K.M., H.B.

## Competing interests

T.K.C reports institutional and/or personal, paid and/or unpaid support for research, advisory boards, consultancy, and/or honoraria past 10 years, ongoing or not, from Alkermes, Arcus Bio, AstraZeneca, Aravive, Aveo, Bayer, Bristol Myers-Squibb, Bicycle Therapeutics, Calithera, Circle Pharma, Deciphera Pharmaceuticals, Eisai, EMD Serono, Exelixis, GlaxoSmithKline, Gilead, HiberCell, IQVA, Infinity, Institut Servier, Ipsen, Jansen, Kanaph, Lilly, Merck, Nikang, Neomorph, Nuscan/PrecedeBio, Novartis, Oncohost, Pfizer, Roche, Sanofi/Aventis, Scholar Rock, Surface Oncology, Takeda, Tempest, Up-To-Date, CME and non-CME events (Mashup Media Peerview, OncLive, MJH, CCO), for work unrelated to the present study. A.K.T. reports institutional research funding from Novartis for work unrelated to the present study. D.E. reports institutional research funding and honoraria from Bayer, Bristol-Myers Squibb, Cardiff Oncology, MiNK Therapeutics, Novartis, Puma Biotechnology, Sanofi, Nimbus Therapeutics, Foundation Medicine, for work unrelated to the present study. S.S. has served as a consultant advisor for Bristol Myers Squibb, Merck Sharp and Dohme, AstraZeneca, Novartis, Janssen and Merck Serono (funds go to the institution) and received research grants for investigator-initiated trials from Novartis, Genentech, Amgen, AstraZeneca, Merck Serono, Merck Sharp and Dohme, Pfizer and Senwha (funds go to the institution) for work unrelated to the present study. H.B. has served as consultant/advisory board member for Astra Zeneca, Merck, Pfizer, Amgen, Novartis, Bayer, Daiichi Sankyo, and has received research funding (to institution) from Janssen, Bristol Myers Squibb, Circle Pharma, Daiichi Sankyo, Novartis for work unrelated to the present study. The remaining authors declare no competing interests.
