## [Transparent Peer Review file · Nature Communications]

Intraindividual epigenetic heterogeneity underlying phenotypic subtypes of advanced prostate cancer

Corresponding Author: Dr Himisha Beltran

Version 0:

Reviewer comments:

Reviewer #1

(Remarks to the Author)

This study investigates the epigenetic heterogeneity of castration-resistant prostate cancer (CRPC) and neuroendocrine prostate cancer (NEPC) by integrating DNA methylation, RNA sequencing, and histone modification profiling across metastatic lesions. The findings reveal DNA methylation-driven gene dysregulation associated with tumor lineage differentiation and potential therapeutic targets, including ASCL1, AR, PSMA, DLL3, STEAP1, and B7-H3. These results provide insights into transcriptional reprogramming mechanisms in CRPC and highlight the potential of multi-omic integration for identifying novel therapeutic strategies.

The following suggestions may help the authors improve their manuscript:

1. Subheadings in the results section would enhance readability and facilitate understanding.
2. Do the clinical samples shown in Figure 1a include follow-up information? Given the valuable data resources in this study, the authors are encouraged to make clinical information and sequencing data publicly available to support future research in this field.
3. The authors have made significant contributions to research on advanced prostate cancer. In Figure 4C, the NE+ category is subdivided into AR+, AR-, and AR-low groups. The authors should clarify the criteria used for this classification and explain its biological significance. Specifically, what role does AR expression play in NE differentiation, and must NE-classified samples be strictly AR-negative?
4. The study identifies several key molecules; are there any in vitro experiments to further validate these findings?

Reviewer #2

(Remarks to the Author)

Tumor heterogeneity across metastatic CRPC tumors has hindered molecular subtyping and the development of targeted therapies for this lethal disease. In this manuscript, Mizuno et al. conducted a multi-omics analysis of metastatic CRPC biopsies collected from multiple anatomic sites within the same patients, with histologically defined adenocarcinoma or NEPC morphology. They found that global methylation and gene expression profiles were conserved across metastatic sites from the same patient. The existing AR/NE signatures classified tumors but also revealed that some patients had tumors falling into more than one category at different metastatic sites. DNA methylation analysis further refined these distinctions, highlighting the epigenetic mechanisms underlying subtype differences. Moreover, DNA methylation-driven gene regulation, based on associations with active or repressive histone modifications (i.e., H3K27ac or H3K27me3, respectively) and their locations in promoters or gene bodies, contributed to the dysregulation of genes involved in prostate lineage determination and therapeutic targeting. Lastly, the authors identified BMP4 signaling as a potential biomarker for double-negative prostate cancer (DNPC, i.e., AR-/NE-). Overall, this study provides valuable insights into CRPC heterogeneity. The following comments may help strengthen its findings:

1. The authors demonstrated that BMP gene expression and BMP signaling signatures were highly upregulated in DNPC. Can the authors provide functional data to examine whether BMP pathway activation promotes malignant phenotypes in DNPC?
2. Can the authors explicitly address the limitations of the sample size and further clarify how these data might be generalized?

3. Can the authors expand the discussion on therapeutic implications, such as the potential for combining epigenetic drugs with immune checkpoint inhibitors or targeted therapies in subsets of DN patients?

4. Subtitles should be provided for each figure in the main text to improve readability and organization.

Reviewer #3

(Remarks to the Author)

In this manuscript, the authors conduct integrative epigenetic profiling across multiple metastatic lesions from patients with CRPC/NEPC. They illustrate how these profiles reveal intraindividual epigenetic heterogeneity and uncover potential mechanisms driving transcriptional reprogramming in CRPC, providing valuable insights into its epigenetic landscape. While the study presents a novel and compelling approach, several aspects require clarification and refinement to strengthen the conclusions and enhance the manuscript's overall impact.

The authors focus specifically on H3K27ac and H3K27me3 modifications. While these histone marks are well-characterized, other modifications may provide additional insights. The rationale for exclusively analyzing these two should be explicitly justified.

The manuscript states that a certified pathologist assessed histology; however, including representative histological images, even in supplementary figures, would strengthen the study by visually validating histological assessments.

The treatment history of the patients is not described. Were treatment regimens similar across individuals? Could variations in treatment contribute to the observed heterogeneity, particularly among outliers? Addressing this aspect is critical for contextualizing the findings.

In the analysis: "To examine this occurrence more thoroughly, the top 10,000 CpG sites with the most variable methylation patterns were utilized to cluster the samples from each patient," it is unclear whether all samples were included. The corresponding figure suggests otherwise. The authors should clarify whether all samples were used, and if not, provide a justification for their selection criteria.

The authors state: "The majority of genes linked with H3K27ac-associated regions showed a negative correlation between their expression and DNA methylation levels. However, 70% of the genes linked with H3K27me3-associated regions exhibited a positive correlation between their expression and methylation levels." However, supporting figures or plots are not provided. These data should be included, at least in the supplementary materials, to substantiate these claims.

The authors report that up to 35% of DN samples exhibited elevated BMP4 expression and pathway activation. However, in the Beltran cohort, this was observed in only 1 out of 5 cases. This discrepancy should be acknowledged, as it impacts the generalizability of the conclusions.

The authors state: "External cohorts of DN tumors with low BMP4 expression also displayed relatively higher immune scores." While this trend is evident in the Beltran cohort, it is less pronounced in the WCDDT cohort. This discrepancy should be acknowledged and discussed to provide a balanced interpretation.

Did the authors assess the prognostic significance of BMP4 expression in any cohort? If not, such an analysis would be highly relevant and could further enhance the clinical implications of the findings.

Given the established role of DNA methylation in regulating immune cell infiltration, did the authors identify specific epigenetic alterations in immune-related genes that could explain the immune-high DN CRPC phenotype observed in this cohort? Exploring this aspect would provide mechanistic insights into the relationship between DNA methylation and immune modulation in CRPC.

By addressing these points, the authors can improve the clarity and robustness of their findings, ultimately increasing the impact of their study.

Version 1:

Reviewer comments:

Reviewer #1

(Remarks to the Author)

The authors' revisions have satisfactorily addressed my concerns, and the manuscript is now suitable for consideration for publication.

Reviewer #2

(Remarks to the Author)

The authors have fully addressed my previous concerns, and the manuscript is ready for publication.

Reviewer #3

(Remarks to the Author)

The authors have adequately addressed the questions and comments raised by this reviewer. Their revisions have significantly improved the clarity of the manuscript and increased the robustness of the findings. The responses were thoughtful and thorough, and the updated version reflects a stronger and more coherent research presentation. I have no further concerns and consider the manuscript suitable for publication pending final editorial review.

Response to Reviewer Comments

Reviewer #1:

This study investigates the epigenetic heterogeneity of castration-resistant prostate cancer (CRPC) and neuroendocrine prostate cancer (NEPC) by integrating DNA methylation, RNA sequencing, and histone modification profiling across metastatic lesions. The findings reveal DNA methylation-driven gene dysregulation associated with tumor lineage differentiation and potential therapeutic targets, including ASCL1, AR, PSMA, DLL3, STEAP1, and B7-H3. These results provide insights into transcriptional reprogramming mechanisms in CRPC and highlight the potential of multi-omic integration for identifying novel therapeutic strategies.

The following suggestions may help the authors improve their manuscript:

1. Subheadings in the results section would enhance readability and facilitate understanding.

Response: We thank the reviewer for this suggestion. We have now added informative subheadings throughout the Results section to improve readability and comprehension of our findings.

Changes to the manuscript: Lines 82, 115, 139, and 197.

2. Do the clinical samples shown in Figure 1a include follow-up information? Given the valuable data resources in this study, the authors are encouraged to make clinical information and sequencing data publicly available to support future research in this field.

Response: We appreciate this suggestion. We have now included additional treatment information for the samples in Supplementary Table 1. All sequencing data (RRBS, RNA-seq, and ChIP-seq / CUT&Tag for H3K27ac and H3K27me3) generated in this study have been deposited in the Gene Expression Omnibus (GEO) public repository and will be publicly available upon acceptance of the manuscript.

3. The authors have made significant contributions to research on advanced prostate cancer. In Figure 4C, the NE+ category is subdivided into AR+, AR-, and AR-low groups. The authors should clarify the criteria used for this classification and explain its biological significance. Specifically, what role does AR expression play in NE differentiation, and must NE-classified samples be strictly AR-negative?

Response: Thank you for this important question regarding the classification criteria for AR/NE categories. In this study, we classified samples based on expression levels of previously defined AR signaling and NEPC signature gene sets (described in Beltran et al,

Nat Med 2016). These gene signatures have been widely applied and published in several datasets including the two SU2C-PCF Dream Team datasets and Bluemn et al. Cancer Cell. 2017. For each sample, AR and NEPC scores were calculated using the previously described methods with the GSVA Bioconductor package in R.

Specifically, we calculated z-scores for each gene in the respective signature sets across all samples. Samples were classified as AR-negative (AR-) if they had an AR score <0 and the sum of AR and KLK3 expression z-scores was <0 ; otherwise, they were classified as AR-positive (AR+ or AR-low). Within AR-positive samples, those with a sum of z-scores for the AR signature gene set >3.5 were classified as AR+, while those with ≤ 3.5 were classified as AR-low. Similarly, samples with a sum of z-scores >0 for the NE1 gene set (as shown in Fig. 2A) were classified as NE+; otherwise, they were NE-.

Regarding the biological significance of AR expression in NE differentiation, our data and previous studies suggest a complex relationship. While most NE+ samples show low or absent AR expression (AR-/NE+), we also identified samples with both AR and NE marker expression (AR+/NE+), sometimes referred to as "amphicrine" tumors. As described by Labrecque et al. (JCI 2019), amphicrine prostate cancer represents tumors composed of cells that co-express AR, PSA, and neuroendocrine markers (CHGA, SYP). This amphicrine state might represent an intermediate phenotype during lineage plasticity though the clinical significance and management considerations are still evolving which we recently described in a Perspective manuscript (Haffner et al, CCR 2024).

Samples that were negative for both AR and NE markers (AR-/NE-) were termed "double-negative" prostate cancer (DNPC). Previously published papers have suggested that DNPC could represent a distinct subset of AR-independent prostate cancer that lacks both AR signaling and neuroendocrine features. These tumors can exhibit alternative lineage programs, including gastrointestinal, squamous, and other phenotypes (Beltran et al. Clin Cancer Res. 2019). Some DNPC tumors demonstrate preferential activation of fibroblast growth factor signaling, suggesting unique biological drivers distinct from both AR-driven and NE-driven disease (Bluemn et al. Cancer Cell. 2017).

This heterogeneity in AR and NE marker expression likely reflects different stages of lineage plasticity and adaptation to treatment pressure. Therefore, NE-classified samples are not required to be strictly AR-negative, as tumors may display various degrees of lineage plasticity along a continuum, with AR+/NE+ representing a potential transitional state between AR+/NE- and AR-/NE+ phenotypes.

We have clarified these classification criteria and their biological implications in the Methods and Results sections of the revised manuscript.

Changes to the manuscript: Lines 120-130 and 450-459.

4. The study identifies several key molecules; are there any in vitro experiments to further validate these findings?

Response: Thank you for this comment. We agree that validation of key molecules and pathways would strengthen the manuscript. To this end, we evaluated the BMP4 signaling pathway and used an inhibitor, LDN-193189, against the BMP4 receptor (ALK2/3) (Response Figure a). Blockade of ALK2/3 in the DU145 DNPC cell line resulted in a significant decrease in cell viability (Response Figure b,c) associated with reduction in phosphorylation of SMAD1/5, the substrate of ALK2/3 (Response Figure d).

We have added these experimental results to the Results section (Fig. 4d,e, Supplementary Fig. 4h,i), expanded our Discussion to include these findings and potential therapeutic implications, and included the detailed methodology in the Methods section.

Changes to the manuscript: Lines 234-245, 324-327, and 528-549.

Reviewer #2:

1. The authors demonstrated that BMP gene expression and BMP signaling signatures were highly upregulated in DNPC. Can the authors provide functional data to examine whether BMP pathway activation promotes malignant phenotypes in DNPC?

Response: Thank you for this comment. We agree that adding functional data to validate the relevance of the BMP pathway in DNPC will strengthen our study. This was also brought up by Reviewer 1. We evaluated the importance of the BMP4 signaling pathway using a pharmacological inhibitor of the BMP4 receptor (ALK2/3), LDN-193189 (LDN), that blocks the activation of BMP signaling (Response Figure a). We used DU145, a representative DNPC cell line, to determine the IC₅₀ of LDN inhibitor to be 2.1 µM (R²=0.99) (Response

Figure b). Using 2 μM of LDN resulted in a dramatic reduction in cell viability (Response Figure c). At this concentration, we observed a notable reduction in phospho-SMAD1/5, the substrate of ALK2/3 (Response Figure d) and a downstream component of the activated BMP4 signaling pathway. Together, these data address the relevance of the activated BMP4 signaling pathway in DNPC proliferation.

We have incorporated these new findings into the Results section (Fig. 4d,e, Supplementary Fig. 4h,i), expanded the Discussion section to highlight potential therapeutic implications and added the detailed experimental procedures to the Methods section.

Changes to the manuscript: Lines 234-245, 324-327, and 528-549.

2. Can the authors explicitly address the limitations of the sample size and further clarify how these data might be generalized?

Response: We thank the Reviewer for this valuable comment regarding sample size limitations and generalizability of our findings. We fully acknowledge the relatively small sample size as a limitation of our study and have emphasized this in the Discussion.

To validate methylation-gene links identified by our integrative regional methylation and gene expression analyses, we used an independent dataset from the West Coast Prostate Cancer Dream Team (WCDT) cohort comprising 128 metastatic CRPC samples with matched RNA-seq and WGBS data. This validation across a substantially larger external dataset significantly strengthens the reliability of our epigenetic regulatory mechanisms identified.

We identified a recurring epigenetic phenomenon in a subset of double-negative prostate cancer (DNPC) samples characterized by DNA methylation-mediated upregulation of the BMP4 signaling pathway. We validated this BMP4 pathway activation not only in our cohort but also in external datasets including the WCDT and our other published cohorts of patient samples, as well as multiple models including patient-derived xenografts (PDXs), cell lines, and organoids (Supplementary Fig. 4b-e), suggesting it represents a biologically significant

and potentially generalizable epigenetic mechanism in DNPC. Similarly, our identification of DNPC samples with high immune scores was corroborated in external cohorts (Supplementary Fig. 4d,e). These consistent observations across diverse sample collections suggest that, despite our limited sample size, we have identified biologically relevant epigenetic mechanisms that appear generalizable to the broader DNPC population.

We acknowledge, however, that as evident from the WCDT cohort validation, not all DNPC samples demonstrate BMP4 pathway upregulation or immune-high phenotype. This heterogeneity within the DNPC subtype underscores the complexity of advanced prostate cancer. We recognize that our findings represent observations from a relatively limited cohort, and further investigation into the epigenetic regulation of gene expression and molecular phenotype transitions in advanced prostate cancer would benefit from larger studies. Our work provides initial insights into these mechanisms and identifies potentially clinically relevant subtypes of DNPC that may guide future research in this field.

To address these limitations directly in the manuscript, we have expanded the Discussion section acknowledging the sample size limitations and elaborating on potential future directions for more comprehensive epigenetic profiling.

Changes to the manuscript: Lines 368-372.

3. Can the authors expand the discussion on therapeutic implications, such as the potential for combining epigenetic drugs with immune checkpoint inhibitors or targeted therapies in subsets of DN patients?

Response: We thank the Reviewer for this excellent suggestion. Our findings supporting DNA methylation changes contributing to molecular subtype transitions and intraindividual heterogeneity suggests a potential utility for DNA methylation-targeted therapies in advanced prostate cancer. Further, the identification of two distinct epigenetically-driven phenotypes within DNPC - one characterized by DNA methylation-mediated upregulation of the BMP4 signaling pathway and another showing DNA methylation-mediated upregulation of genes associated with immune processes - provides potential rationale for developing combination strategies. For DNPC tumors exhibiting BMP4 pathway activation, combination strategies targeting DNA methylation and BMP4 signaling could be explored.

The DNPC tumors with immune-high profiles identified in our study might be particularly responsive to combinations of DNA methyltransferase inhibitors with immune checkpoint inhibitors. Recent clinical trials in other cancer types have demonstrated the potential of such approaches. For example, Gojo et al. (Blood, 2019) reported promising activity with azacitidine plus pembrolizumab in newly diagnosed older AML patients, while Noviello et al.

(Nat Commun, 2023) documented long-term benefits with guadecitabine plus ipilimumab in advanced melanoma. These findings provide a compelling rationale for further exploration in the immune-high DNPC subset.

We have expanded the Discussion section to address the potential therapeutic implications of our findings.

Changes to the manuscript: Lines 327-329 and 343-348.

4. Subtitles should be provided for each figure in the main text to improve readability and organization.

Response: We have now added descriptive subtitles for each figure in the main text to improve readability and help readers navigate through our findings more efficiently.

Reviewer #3:

The authors focus specifically on H3K27ac and H3K27me3 modifications. While these histone marks are well-characterized, other modifications may provide additional insights. The rationale for exclusively analyzing these two should be explicitly justified.

Response: We appreciate the Reviewer's insightful comment. Indeed, while multiple histone modifications contribute to epigenetic regulation, we could not perform all marks and strategically selected H3K27ac and H3K27me3 based on their specific relevance to our research questions and the biology of prostate cancer.

Our rationale for focusing on these two particular histone marks was multi-faceted. H3K27ac serves as a well-established marker of active enhancers and promoters, while H3K27me3 represents a repressive mark associated with gene silencing. This complementary pair allows us to simultaneously investigate both activating and repressive epigenetic mechanisms within the same genomic contexts and integrate with matched DNA methylation and transcriptome data. While other activating marks such as H3K4me1 and H3K4me3 have more restricted distributions (H3K4me1 primarily at enhancers and H3K4me3 at promoters), H3K27ac is uniquely positioned at both active enhancers and promoters. This broader distribution allows us to examine DNA methylation correlations with gene expression across diverse regulatory elements beyond promoter regions alone.

In contrast to H3K9me3, which predominantly enriches at gene-poor tandem repeat structures, H3K27me3 localizes to gene-rich regions, making it more suitable for our integrated analysis of DNA methylation and gene expression correlations in functionally

relevant genomic contexts. Furthermore, the mechanistic link between EZH2 (the methyltransferase catalyzing H3K27 trimethylation) and lineage plasticity in prostate cancer has been well-documented in the literature.

Technical limitations in tissue quantity from clinical samples necessitated prioritization of the most informative histone marks, as metastatic tissue specimens are often limited in size and quantity. Given these constraints, we selected modifications with established protocols and reliability in clinical specimens that would provide the most comprehensive insights into the regulatory mechanisms underlying phenotypic heterogeneity.

We have now expanded the Discussion to address this point.

Changes to the manuscript: Lines 372-376

The manuscript states that a certified pathologist assessed histology; however, including representative histological images, even in supplementary figures, would strengthen the study by visually validating histological assessments.

Response: We thank the Reviewer for this valuable suggestion. We have now included representative histological images from our patient samples in a new supplementary figure (Supplementary Fig. 5). These images provide visual context for the samples analyzed in our study.

Changes to the manuscript: Lines 393-394.

The treatment history of the patients is not described. Were treatment regimens similar across individuals? Could variations in treatment contribute to the observed heterogeneity, particularly among outliers? Addressing this aspect is critical for contextualizing the findings.

Response: We thank the Reviewer for highlighting this important point. We have now included prior systemic therapies (those administered before autopsy or biopsy sample collection) in an expanded Supplementary Table 1. Across our cohort, treatment regimens were relatively consistent, with most patients receiving standard-of-care therapies for advanced prostate cancer, including androgen deprivation therapy, androgen receptor signaling inhibitors, and taxane-based chemotherapy. For patients with NEPC, platinum-based chemotherapy in combination with etoposide was commonly administered, consistent with standard clinical practice. Overall, we observed limited variation in the types of systemic therapies received by patients in our cohort.

Regarding the potential influence of treatment history on intraindividual heterogeneity, we observed no clear association between treatment history and the presence of heterogeneous phenotypes. While patient WCM63, who exhibited intraindividual heterogeneity, had received eight different systemic therapies prior to autopsy, other cases with heterogeneous phenotypes (CA0090, DFCI24, WCM3672) had undergone only 2-3 prior systemic therapy. This suggests that the extent of treatment exposure alone does not fully explain the observed intraindividual heterogeneity in our cohort. Rather, as our data indicate, intrinsic tumor biology and epigenetic mechanisms likely play more significant roles in driving phenotypic heterogeneity.

We have also addressed this point in the Discussion section.

Changes to the manuscript: Lines 353-365

In the analysis: "To examine this occurrence more thoroughly, the top 10,000 CpG sites with the most variable methylation patterns were utilized to cluster the samples from each patient," it is unclear whether all samples were included. The corresponding figure suggests otherwise. The authors should clarify whether all samples were used, and if not, provide a justification for their selection criteria.

Response: We appreciate the Reviewer's careful reading of our manuscript and the opportunity to clarify this important methodological point. For the analysis described in the statement "To examine this occurrence more thoroughly, the top 10,000 CpG sites with the most variable methylation patterns were utilized to cluster the samples from each patient," we indeed focused specifically on the five cases that exhibited intraindividual heterogeneity (CA0090, DFCI24, DFCI53, WCM3672, WCM63). For each of these five patients, we included all samples from that individual and performed clustering based on the top 10,000 most variable CpG sites within each patient's sample set (not across all patients). This patient-specific approach was intentionally chosen to better capture the methylation patterns that distinguish different molecular subtypes within the same individual. This methodology is described in the Methods section under "DNA Methylation Analysis of Specific Cases with Intraindividual Heterogeneity", but we acknowledge that this connection could be made clearer in the main text.

We have now revised the manuscript to explicitly state this approach in the Results section to avoid any confusion.

Changes to the manuscript: Lines 131-135

The authors state: "The majority of genes linked with H3K27ac-associated regions showed a negative correlation between their expression and DNA methylation levels. However, 70% of the genes linked with H3K27me3-associated regions exhibited a positive correlation between their expression and methylation levels." However, supporting figures or plots are not provided. These data should be included, at least in the supplementary materials, to substantiate these claims.

Response: We thank the Reviewer for the thorough assessment of our manuscript. In Figure 3b, we show the distribution of correlations between DNA methylation and gene expression categorized by genomic context (H3K27ac-associated regions, H3K27me3-associated regions, promoters, and gene bodies). As noted in the figure legend, blue text indicates negative correlation between DNA methylation and gene expression, while red text indicates positive correlation.

The numerical values are also displayed in Fig. 3b and show that for H3K27ac-associated regions, 1,640 genes demonstrated negative correlation between expression and DNA methylation levels (blue text), while 328 genes showed positive correlation (red text). This supports our statement that "the majority of genes linked with H3K27ac-associated regions showed a negative correlation between their expression and DNA methylation levels."

For H3K27me3-associated regions, 238 genes demonstrated negative correlation between expression and DNA methylation levels (blue text), while 507 genes showed positive correlation (red text). This corresponded to approximately 70% (507 out of 745 total genes) showing positive correlation, supporting our statement that "70% of the genes linked with H3K27me3-associated regions exhibited a positive correlation between their expression and methylation levels."

We have now added clearer cross-references in the revised manuscript to Figure 3b to ensure readers can easily locate the supporting data for these statements.

Changes to the manuscript: Lines 158 and 160.

The authors report that up to 35% of DN samples exhibited elevated BMP4 expression and pathway activation. However, in the Beltran cohort, this was observed in only 1 out of 5 cases. This discrepancy should be acknowledged, as it impacts the generalizability of the conclusions.

Response: We thank the Reviewer for this important observation regarding BMP4 pathway activation in DN CRPC samples. The 35% refers to the combined analysis of DN CRPC samples from both the WCDT cohort (13 samples) and the Beltran cohort (4 samples),

where we observed BMP4 pathway activation in a total of 6 samples out of 17 (35%). Specifically, this included 5 samples from the WCDT cohort and 1 sample from the Beltran cohort. We acknowledge that this distinction was not clearly presented in our original manuscript.

We agree with the Reviewer that there is a difference in the proportion of BMP4-activated samples between these cohorts (5/13 in WCDT versus 1/4 in Beltran). The varying proportions across cohorts further emphasize the heterogeneity within DN CRPC and underscore the need for additional studies with larger sample sizes to better characterize the molecular subtypes within this CRPC variant.

We have clarified these points in the Results section of the revised manuscript.

Changes to the manuscript: Lines 223-226.

The authors state: "External cohorts of DN tumors with low BMP4 expression also displayed relatively higher immune scores." While this trend is evident in the Beltran cohort, it is less pronounced in the WCDT cohort. This discrepancy should be acknowledged and discussed to provide a balanced interpretation.

Response: We thank the Reviewer for this insightful observation. We agree that while the trend of DN tumors with low BMP4 expression exhibiting higher immune scores is evident in our cohort and the Beltran cohort, this pattern is less consistent in the WCDT cohort, where some samples show both low BMP4 expression and low immune scores. This important discrepancy warrants discussion, and we have revised our manuscript to provide a more balanced interpretation. Additionally, we have revised our interpretation to acknowledge this heterogeneity, which highlights the likely existence of additional molecular drivers beyond BMP4 and immune pathways in DN CRPC.

Changes to the manuscript: Lines 265-269.

Did the authors assess the prognostic significance of BMP4 expression in any cohort? If not, such an analysis would be highly relevant and could further enhance the clinical implications of the findings.

Response: We thank the Reviewer for this insightful question regarding the prognostic significance of BMP4 expression in prostate cancer. We agree that exploring this relationship would strengthen the clinical relevance of our findings.

To address this comment, we performed an additional analysis using the West Coast Prostate Cancer Dream Team (WCDT) metastatic castration-resistant prostate cancer (mCRPC) cohort, examining the relationship between BMP4 expression and patient outcomes. We categorized patients into high expression (TPM > 10, n=13) and low expression (TPM ≤ 10, n=84) groups and analyzed overall survival from CRPC diagnosis using Kaplan-Meier methodology.

Our analysis revealed no statistically significant difference in overall survival between patients with high versus low BMP4 expression (p=0.32, Response Figure). However, we acknowledge that this assessment is constrained by the relatively small sample size, particularly in the high expression group, which limits statistical power. Furthermore, the heterogeneity of prior treatments and disease characteristics in this retrospective cohort may confound the relationship between BMP4 expression and clinical outcomes.

These findings suggest that while BMP4 expression may serve as an important molecular marker distinguishing double-negative CRPC subtypes as demonstrated in our main analyses, its independent prognostic value requires further validation in larger, prospectively designed studies.

We have added this analysis to the Results section of our revised manuscript (Supplementary Fig. 4j) and included a brief discussion of these findings in the Discussion section, highlighting both the results and limitations of this prognostic assessment, along with emphasizing the need for larger prospective studies to definitively establish the prognostic significance of BMP4 expression in advanced prostate cancer.

Changes to the manuscript: Lines 246-251 and 329-334.

Given the established role of DNA methylation in regulating immune cell infiltration, did the authors identify specific epigenetic alterations in immune-related genes that could explain the immune-high DN CRPC phenotype observed in this cohort? Exploring this aspect would provide mechanistic insights into the relationship between DNA methylation and immune modulation in CRPC.

We thank the Reviewer for this insightful question regarding the relationship between DNA methylation and immune modulation in CRPC. To address this point, we performed additional analyses focusing on the immune-high DN CRPC samples in our cohort (WCM3672 and CA0090).

Our analysis revealed 291 and 288 DNA methylation-upregulated genes in these DN samples, respectively, with 94 genes common between both samples (Response Figure). To explore potential immune-related functions of these common genes, we conducted Gene Ontology (GO) analysis. As expected, we observed significant enrichment of immune process-related GO terms among these 94 genes. The Response Table presents the top 10 enriched GO terms. Among the common methylation-upregulated genes, we identified several immunologically relevant genes including *VAV1*, *IL4R*, *CD300A*, *ITGB2*, and *CARD11*. These genes are known to be involved in critical immune functions such as T cell activation, regulatory T cell suppression, and lymphocyte migration.

The upregulation of these immune-related genes associated with altered DNA methylation patterns provides a potential mechanistic link between epigenetic modifications and the immune-high phenotype observed in these DN CRPC samples. These epigenetic alterations may contribute to immune cell infiltration and activity within the tumor microenvironment of DN CRPC, providing a foundation for future investigations into the epigenetic regulation of tumor-immune interactions in advanced prostate cancer.

We have incorporated these findings into the Results section of our revised manuscript, with the data presented in Supplementary Fig. 4m and Supplementary Table 3.

DNA methylation-upregulated genes
in DN samples

ID	Description	Count	p.adjust	Genes
GO:0007159	leukocyte cell-cell adhesion	23	2.86E-15	VAV1, L4R, CD300A, IITGB2, CARD11
GO:0002768	immune response-regulating cell surface receptor signaling pathway	21	1.54E-14	VAV1, CD300A, CARD11
GO:0002429	immune response-activating cell surface receptor signaling pathway	18	5.86E-12	VAV1, CD300A, CARD11
GO:1903037	regulation of leukocyte cell-cell adhesion	19	5.86E-12	VAV1, IL4R, CD300A, ITGB2, CARD11
GO:0030098	lymphocyte differentiation	19	2.09E-11	VAV1, IL4R, CARD11
GO:0043299	leukocyte degranulation	11	2.68E-11	IL4R, ITGB2
GO:1903039	positive regulation of leukocyte cell-cell adhesion	16	5.04E-11	VAV1, IL4R, ITGB2, CARD11
GO:0002274	myeloid leukocyte activation	15	8.96E-11	IL4R, ITGB2
GO:0002443	leukocyte mediated immunity	19	1.17E-10	VAV1, IL4R, ITGB2
GO:0030217	T cell differentiation	16	2.20E-10	VAV1, IL4R, CARD11

Changes to the manuscript: Lines 256-262.

We thank the reviewers again for their valuable feedback, which has significantly improved our manuscript. We hope that our revisions adequately address their concerns.